

# The regulation of hsacirc_004413 promotes proliferation and drug resistance of gastric cancer cells by acting as a competing endogenous RNA for miR-145-5p

Fusheng Zhou[1,*], Weiqun Ding[2,*], Qiqi Mao[1], Xiaoyun Jiang[1], Jiajie Chen[1], Xianguang Zhao[1], Weijia Xu[1], Jiaxin Huang[1], Liang Zhong[1] and Xu Sun[1]

[1] Department of Gastroenterology, Huashan Hospital North, Fudan University, Shanghai, China
[2] Department of Gastroenterology, Huashan Hospital, Fudan University, Shanghai, China
* These authors contributed equally to this work.

## ABSTRACT

**Background:** Whether circRAN, which acts as a microRNA sponge, plays a role in 5-fluorouracil (5-Fu) resistant gastric cancer has not been reported. In this study, a 5-Fu resistant cell line with an IC50 of 16.59 μM was constructed.
**Methods:** Using comparative analysis of circRNA in the transcriptomics of resistant and sensitive strains, 31 differentially expressed circRNAs were detected, and the microRNA interacting with them was predicted.
**Results:** Hsacirc_004413 was selected for verification in drug resistant and sensitive cells. By interfering with hsacirc_004413 using antisense RNA, the sensitivity of drug resistant cells to 5-Fu was significantly promoted, and the apoptosis and necrosis of the cells were significantly increased. In sensitive cells, inhibition by inhibitors enhanced the resistance of cells to 5-Fu. We hypothesize that hsacirc_004413 makes gastric cancer cells resistant to 5-Fu mainly through adsorption of miR-145-5p.

## INTRODUCTION

Gastric carcinoma (GC) is the third leading cause of cancer-related death worldwide (*Bray et al., 2018*), especially in China and other Southeast Asian countries (*Forman & Burley, 2006*). At present, in addition to surgery, immunotherapy, chemotherapy is the main clinical treatment for GC patients (*Kelly, 2017*; *Zhang et al., 2011*). The 5-fluorouracil (5-Fu) has been approved by the Federal Drug Administration for the treatment of GC (*Sakuramoto et al., 2007*). As first-line therapy in patients with gastric cancer, 5-Fu showed encouraging antitumor activity in combination therapy with targeted drugs or monoclonal antibodies (*Xia et al., 2021*; *Hosoda et al., 2020*; *Bang et al., 2017*). However, many patients will develop drug resistance after using 5-Fu, which largely limits the efficacy of chemotherapy in patients (*Borst & Elferink, 2002*). Therefore, studying the mechanism of 5-Fu resistance in GC cells is necessary to improve the therapeutic effect of 5-Fu in patients and to prolong their survival times.

Corresponding author
Xu Sun, sunxu6060@163.com

The circular RNAs (circRNAs) comprise is a class of non-coding RNA molecules with a closed circular structure, without a 5′cap and 3′poiy(A) structure, mainly located in the cytoplasm or stored in exosomes, unaffected by RNA exonuclides. They have been shown to exist in a variety of eukaryotes (*Li et al., 2015*; *Kristensen et al., 2019*). A large number of studies have shown that circRNAs are closely related to the growth and development of organisms, the stress response, disease occurrence and development (*Kristensen et al., 2019*; *Goodall & Wickramasinghe, 2021*; *Barbieri & Kouzarides, 2020*). Currently, the biological functions of circRNAs that are widely recognized, and mainly include circRNAs as miRNA sponges and microRNA (miRNA) binding sites. In addition, the inhibition of miRNA on target genes was removed. CircRNAs can bind to specific proteins and play a regulatory role by inhibiting or cooperating with the function of proteins. In addition, circRNA-derived peptides that circRNA serve as protein templates to produce, they also be expressed under different conditions to act as intracellular regulatory protein products (*Li et al., 2015*; *Kristensen et al., 2019*; *Goodall & Wickramasinghe, 2021*; *Barbieri & Kouzarides, 2020*). There is increasing evidence that microRNAs (miRNAs), which are significantly different in normal and drug resistant tumor tissues (*He & Hannon, 2004*; *Wen et al., 2020*; *Si et al., 2021*; *Zhao et al., 2021*), play important roles in the regulation of gene expression, including cell proliferation, apoptosis, tumorigenesis, and multidrug resistance of cancers (*Zheng et al., 2017*; *Marjaneh et al., 2019*; *Moradi-Marjaneh et al., 2019*; *Shi et al., 2015*). The circRNA with the role of a "sponge" of miRNAs, usually competitively binds miRNAs *in vivo*, thus reducing the role of miRNAs in the regulation of gene expression (*Hansen et al., 2013*; *Salmena et al., 2011*). However, there are currently few reports on the role of circ RNAs in chemotherapy resistance.

In the present study, drug resistant cell lines of SGC-7901-5-Fu were established through screening of drug resistant cell lines to identify circRNA differences between SGC-7901-and SGC 7901-5-Fu cells. In addition, two circRNA were selected to identify miRNAs that could bind to each other, which were verified by overexpression and interference.

## MATERIALS AND METHODS

### Cell culture and drug resistant cell construction

Human GC tumor cell lines SGC-7901 and MKN-45 were purchased from iCell Bioscience (Shanghai, China). The 5-Fu-sensitive gastric cancer line, SGC-7901, was cultured in RPMI 1640 medium (Gibco, Gaithersburg, MD, USA) containing 10% fetal bovine serum (Gibco, Gaithersburg, MD, USA) and 1% penicillin-streptomycin (Gibco, Gaithersburg, MD, USA) at 37 °C in a 5% $CO_2$ incubator. The cells were digested with 0.25% trypsin and routinely passaged when the cells grew to 70–80% confluency. The stepwise addition method was used, starting from addition of 10 nmol/L 5-Fu (Yuanye Bio-Technology, Shanghai, China) until the cells can grew stably in culture medium containing 10 µM 5-Fu; the constructed drug-resistant cell line was named SGC-7901-5-Fu. The drugs were withdrawn 2 weeks before the experiment and routinely cultivated for future use.

## Detecting cell proliferation using CCK-8 assay

SGC-7901 and SGC-7901-5-Fu cells in the logarithmic growth phase were seeded in 96-well plates ($5 \times 10^4$ cells/mL) with three replicate wells in each group. After the cells adhered on the second day, the two kinds of cells were treated with different concentrations of 5-Fu (0.1, 1, 5, 10, 20, and 50 μmol/L), and 10 μL/well CCK-8 assay reagent was added after 48 h. After culturing the cells for 2 h, a microplate reader was used to measure the absorbance of each well at 450 nm. The half maximal inhibitory concentration ($IC_{50}$) value of 5-Fu and the drug resistance indices of the two cell lines were then calculated.

## RNA library construction and RNA-seq sequencing

The Illumina RNA-Seq procedure (Illumina, San Diego, CA, USA) for RNA library construction was conducted by Nanjing Pisenuo Gene Technology (Nanjing, China). The standard for library construction was RNA integrity number (RIN) ≥ 7, 28 S/18 S ratio > 0.7. The HiSeqTM 2,500 sequencer (Illumina, San Diego, CA, USA) was then used for sequencing with a sample load of 1 μg. The sequencing results were compared and annotated with the database as analysis background data, and the screening conditions for differential expression of circRNA in the two cells line were defined as fold changes (FC) ≥ 2 and $P < 0.05$.

Analysis of GO enrichment and KEGG pathway analysis of genes derived from the differential circRNAUse GO (Gene Ontology) database (http://www.geneontology.org) was used to perform gene function enrichment analysis on genes derived from differential circRNA, then count the numbers of differential genes included in each GO entry were counted, and the hypergeometric distribution test method was used to calculate the significance of the enrichment of differential transcripts in each GO entry. KEGG (Kyoto Encyclopedia of Genes and Genomes) database (http://www.genome.jp/kegg/pathway.html) was used to analyze the signal pathways (Pathway) of the differential genes, and also used of hypergeometric distribution test to calculatethe significances of the enrichment of differential genes in each pathway entry. The above calculations resulted in a value of P that reflected the significance of enrichment. The smaller the $P$ value, the more significant the tendency of differential genes to be enriched in the GO or Pathway entry.

## Real-time quantitative PCR to verify differential circRNA levels

Total RNA was extracted from SGC7901 and SGC7901-5-Fu cells using TRIzol reagent (TaKaRa, Dalian, China). The RNA concentration was measured using a NanoDrop spectrophotometer (Thermo Fisher Scientific, Waltham, MA, USA), then RNA was reverse transcribed into cDNA using a reverse transcription kit (TaKaRa, Dalian, China), and the cDNA was used as a template for real-time fluorescence quantitative PCR. The upstream primer sequence of hsacirc_004413 was 5′-AGCTGCTCAGAGAGACACAG-3′, the downstream primer sequence was 5′-TCCTCCCATCAAGCCCATTT-3′; the upstream primer sequence of β-actin gene was 5-GATCCACATCTGCTGGAAGG-3′, and the downstream primer sequence is 5-AAGTGTGACGTT GACATCCG-3′. The PCR reaction

conditions: were the following: predenaturation at 95 °C for 30 s; denaturation at 95 °C for 5 s, and annealing at 60 °C for 30 s, followed b35 cycles.

## Transfection of antisense RNA

The cells were overgrown to 80–90% confluency and transfected. Before transfection, the old culture medium was removed and the cells were washed three times with serum-free culture medium, then 450 µL of serum-free culture medium was added to each well in a 24-well plate. Two µLof Lipofectamine 2,000 was then diluted with 50 µL of OpTI-MEM (Thermo Fisher Scientific, Waltham, MA, USA). Then the antisense RNA (Table S1, synthesized by Gene-Pharma, Suzhou, China) was diluted with 50 µL of opti-MEM and filled with a 1 µg mixture containing 50 µL of the two diluents and incubated at room temperature for 10 min. The 50 µL mixture was then added to each well and placed in an incubator,and was cultured for 24 h for other experiments.

## Apoptosis and necrosis staining

After cell transfection, 5 µL of 4′, 6-diamidino-2-phenylindole (DAPI) staining solution was added to each well and incubated at room temperature for 10 min, followed by addition of 5 µL of propidium iodide (PI) staining solution for 10 min at room temperature. After washing with phosphate-buffered saline three times, blue and red fluorescence were observed using fluorescence microscope.

## Statistical analysis

We used SPSS 17.0 statistical software for Windows for statistical analyses (SPSS, Chicago, IL, USA). Prism 5.0 software (GraphPad, San Diego, CA, USA) were used for the design of hSACIRC_004413 primers. analyses. The difference of data between control group and experimental group and different experimental groups was analyzed using with Student's t-test or one-way analysis of variance by SPSS 17.0 statistical software. Data are expressed as the mean+−standard deviation (SD). The results were considered statistically significant at a value of $P < 0.05$.

# RESULTS

## Establishment of 5-Fu resistant cell lines in GC

The 5-Fu resistant gastric cancer cell line, SGC-7901-5-Fu, was established by low concentration gradient increments, combined with high concentration intermittent shock *in vitro*, and was named SGC-7901-5-Fu. The proliferation inhibition rates of SGC-7901 and drug resistant cells were detected using the CCK-8 assay. The results showed that the proliferation inhibition of SGC7901 and drug-resistant cells increased gradually with an increase of 5-Fu concentration. Prism5 software (GraphPad, San Diego, CA, USA) was used to calculate the $IC_{50}$ of both types of cells. The results show the $IC_{50}$ of drug resistant cells (Fig. 1B) and SGC-7901 cells (Fig. 1A), which was significantly higher than that of the parent cells (>5×).

Differentiation of circRNA expression profiles between SGC-7901 and SGC-7901-5-Fu cells. Next, the circRNA expression profiles of SGC-7901 and SGC-7901-5-Fu cells were

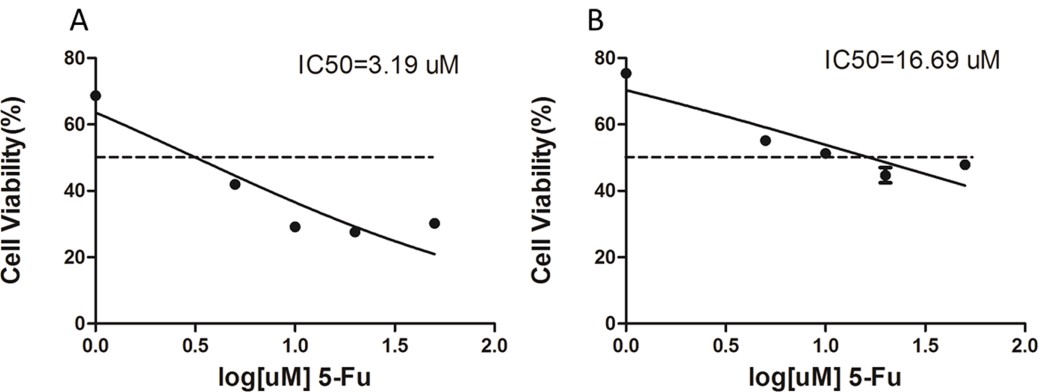

**Figure 1** The IC$_{50}$ value for 5-fluorouracil (5-Fu) resistant gastric cancer cells. Cell lines resistant to 5-F were constructed by *in vitro* addition of low concentration increments combined with high concentration intermittent shock. (A) The IC$_{50}$ of the SCG-7901-5-Fu drug-resistant strain, (B) the IC$_{50}$ of the SCG-7901 parental strain.

analyzed using a high-throughput RNA-seq technology. After normalization, a total of 4,496 circRNA targets were found in SGC-7901 cells (Table S2). As a result, 31 distinct circRNAs were differentially expressed between SGC-7901 and SGC-7901-5-Fu cell lines with a fold change >2.0, and a value of $P < 0.05$, of which 10 were upregulated and 21 were downregulated (Table S3). These up-and downregulated circRNAs are shown in the heat map in Fig. 2B.

## The hsacirc_004413 was highly expressed in 5-Fu resistant SGC-7901-5-Fu cells

First, we determined the expression levels of hsacirc_004413 in GC cell lines (MKN-45, SGC-7901, and SGC7901-5-Fu), and compared with MKN-45, the poorly-differentiated human gastric cancer cells, SGC-7901, and the moderately-differentiated human gastric cancer cells, and SGC-7901-5-Fu cell expressions were increased. Figure 3 shows that the expression levels of hsacirc_004413 in MKN-45 and SGC-7901 cells were significantly lower than that of the drug-resistant SGC-7901-5Fu cells. These results indicated that the increased expression of hsacirc_004413 in SGC-7901-5-Fu cells may be involved in the resistance of gastric cancer cells to 5-Fu.

## The hsacirc_004413 knockdown enhanced 5-Fu sensitivity in SGC-7901-5-Fu cells

To confirm the role of hsacirc_004413 in resistance to 5-Fu in SGC-7901-5-Fu cells, antisense RNA with knockdown activity was synthesized. After 24 h of transfection, we investigated the expression levels of hsacirc_004413 in SGC-7901 and SGC-7901-5-fu cells (Figs. 4A, 4B). After knockdown, the expression level was only about 60% that of the non-knocked down group, which indicated that knockdown was successful. There was no significant difference in SGC-7901 cells. As-NC as a negative control group did not affect the expression of hsacirc_004413 in both these cell groups. Then, 10 μM 5-Fu was added to the medium to determine the survival percentage of cells transfected with

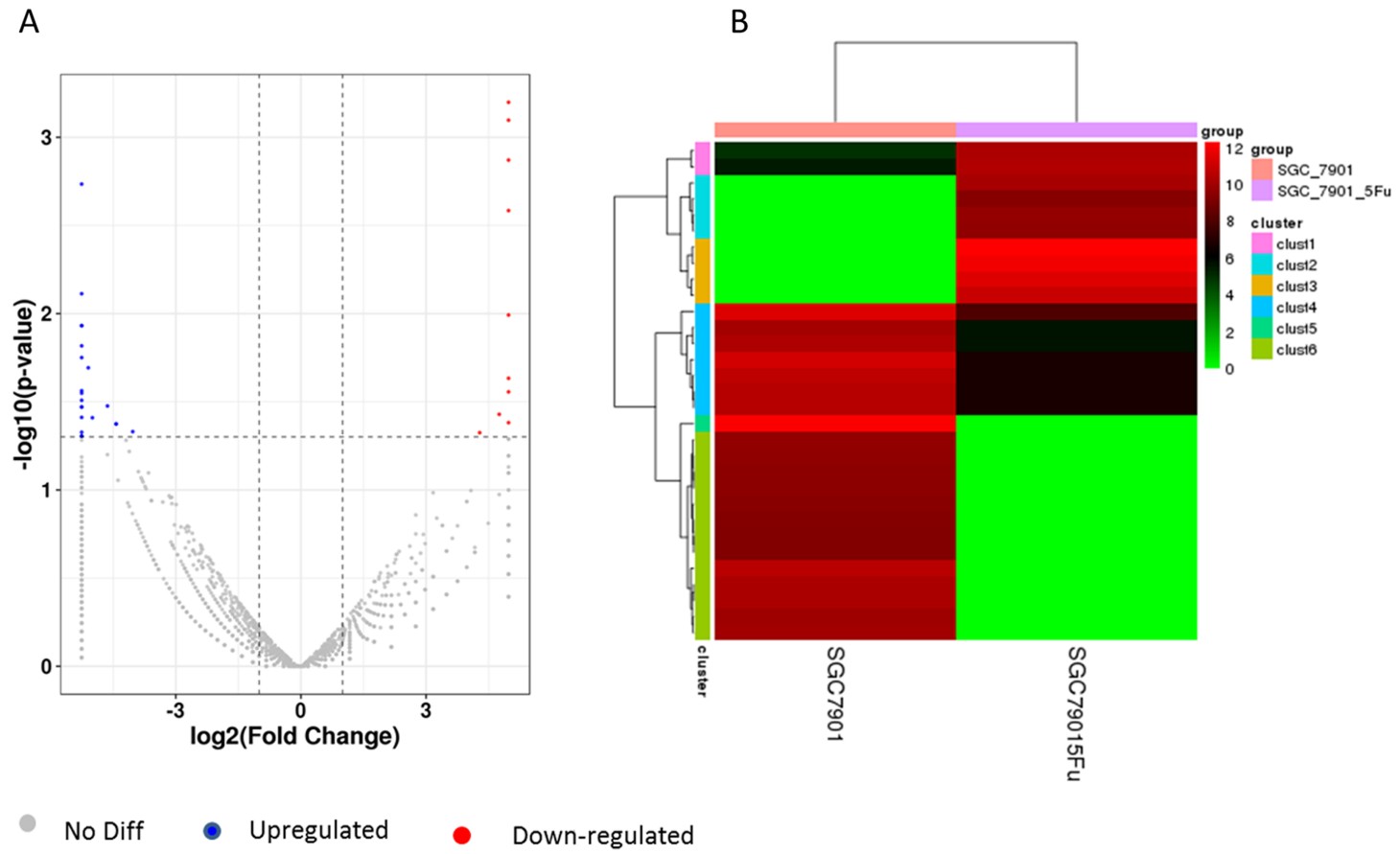

**Figure 2 The volcano plot and cluster diagram of circRNA differentially expressed in SGC-7901-5-Fu cells.** (A) The x-coordinate is the $\log_2$ fold change and the y-coordinate is the $-\log_{10}$ (P-value). The two vertical dotted lines in the figure are two times that of the difference threshold. The dotted line is the p-value = 0.05 of the threshold. Red dots indicate upregulated circRNA, blue dots indicate downregulated circRNA, and gray dots indicate insignificantly differentially expressed circRNAs. (B) CircRNA laterally, with one sample for each row; red represents circRNA with high expression, and green represents circRNA with low expression.

antisense RNA. The results are shown in Figs. 4C, 4D. After transfection, the cell survival percentage was significantly lower than that of the non-knockdown group and the negative control group. However, no such process was found in SGC-7901 cells. The results suggested that hsacirc_004413 played an important role in the drug resistance mechanism of SGC-7901-5-Fu cells.

The 5-Fu mainly inhibits cell mitosis, which leads to cell apoptosis and necrosis. After hsacirc_004413 treatment, cell apoptosis and necrosis were detected. Figure 5 shows that at the same concentration of 5-Fu, apoptosis (solid line arrow) and necrosis (dotted line arrow) were observed in the sensitive SGC-7901 strain, 48 h later, in which apoptotic cells accounted for the majority of changes and necrotic cells accounted for only a small percentage. The percentages of apoptotic and necrotic cells in drug-resistant strains of SGC-7901-5-Fu were lower than that in sensitive strains. After transfection of drug-resistant strains with siRNA, the percentages of cell apoptosis and necrosis increased significantly. Figure 5 shows that the cells of drug-resistant strains became more sensitive

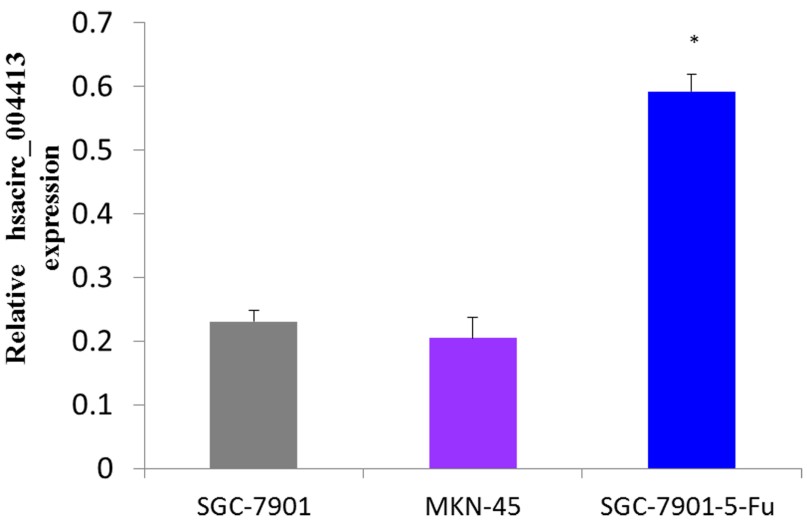

**Figure 3 The relative RNA levels of hsacirc_004413 in SGC-7901-5-fu, SGC-7901 and MNK45 gastric cancer cells were determined by qPCR.** SGC-7901 is a moderately differentiated gastric cancer cell; MNK45 is a poorly differentiated gastric cancer cell; SGC-7901-5-fu cell lines resistant to 5-Fu were obtained by low concentration increase in vitro combined with high concentration intermittent rest method. $^*P < 0.05$ indicates significant differences.

to drugs after interference, consistent with the possibility that hsacirc_004413 played an important role in the resistance mechanism of 5-Fu.

## Inhibition of miR-145-5p improved the resistance of sensitive GC cells to 5-Fu

We used bioinformatics to predict microRNAs that may have interact with hsacirc_004413 (Table S5). The miR-145-5p has been shown to inhibit the proliferation of tumor cells in GC by promoting cell apoptosis and increasing sensitivity of tumor cells to chemotherapeutic agents in chronic myeloid leukemia patients (Zeng et al., 2017). To confirm our prediction, we obtained an inhibitor of miR-145-5p (synthesized by Gene-Pharma) and transfected it into the cells. After 24 h, the CCK8 assay was used to detect the survival percentage of cells. The results are shown in Fig. 6. After addition of 5-Fu and NC inhibitors, the cell survival percentage decreased, and there was no restriction difference compared with the addition of 5-Fu only. The cells that were added to 5-Fu and inhibitors showed a significant increase in survival. The survival rate was close to that of the SGC-7901-5-Fu drug resistant cells. After addition of the miR-145-5P inhibitor, the expressions of related genes were inhibited because circRNA adsorbed microRNA and inhibited its effect. We therefore hypothesized that hsacirc_004413 was rich in miR-145-5p binding sites to act as miRNA sponges, preventing miR-145-5p from interacting with mRNAs in the 3′ untranslated region, which may have been involved in inhibiting cell apoptosis. These data suggested that hsacirc_004413 inhibited apoptosis by sponging miR-145-5p, thereby promoting SGC-7901-5-Fu cell proliferation, and chemoresistance.

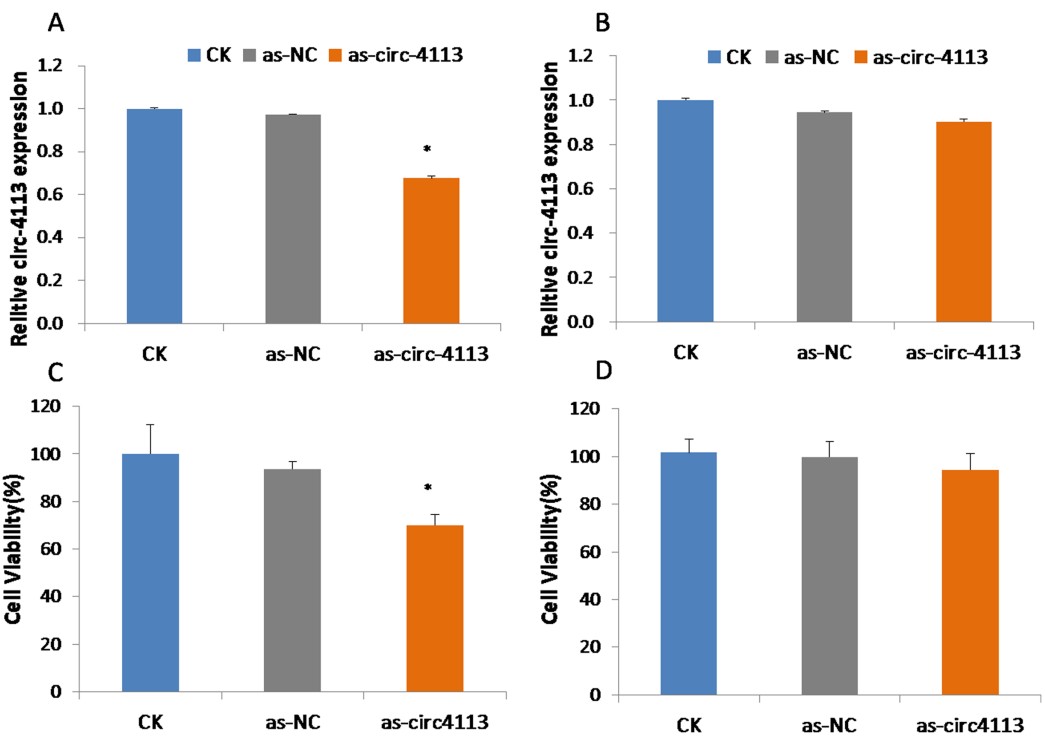

**Figure 4 SGC-7901-5-fu and SGC-7901 cells were transfected with antisense RNA targeting hsacirc_004413 or NC siRNAs (as-NC) for 24 h.** Cells survival percentage was examined by CCK8. (A) The expression of hsacirc_004413 in SGC-7901-5-Fu cells after knockdown. (C) The cell survival percentage. (B) The expression of hsacirc_004413 in SGC-7901 cells after knockdown. (D) The cell survival percentage. *$P < 0.05$ indicates significant differences.

## DISCUSSION

It has been reported that miR-185, miR-218, miR-145, miR-27b, miR-30a, miR-107, miR-Bart20-5p, and miR-23b-3p are involved in drug resistance of 5-Fu in GC (*Chen et al., 2019*). CircRNA is highly expressed in the testes (*Memczak et al., 2013*), which is a sponge for the expressed mirRNA to regulate downstream genes (*Chen et al., 2019*; *Ebert & Sharp, 2010*; *Capel et al., 1993*; *Wilusz & Sharp, 2013*). It has therefore been verified that circRNAs take part in many kinds of cancer, particularly in the context of some drug resistance, such as for circ-pvt1, in promoting pacaxel resistance in GC (*Li et al., 2015*; *Shao et al., 2017*).

Due to the heterogeneity of tumor tissue, we constructed a drug resistant strain of SGC-7901-5-Fu with an IC50 of 16.59 μM by using SGC-7901 parental cells with *in vitro* low concentration increments combined with high concentration intermittent shock. The two transcriptomics were compared to find circRAN with different expressions. Figure 2 shows that there were a total of 31 differentially expressed genes, 10 upregulated genes, and 21 downregulated genes. As an analogue of uracil, 5-Fu enters the human body through the blood and participates in the synthesis of uracil and thymidine. Due to the insertion of a fluorine atom at position 5 of the heterocyclic ring, its spatial structure changes, so it can inhibit thymidine synthase and hinder the synthesis of pyrimidine and

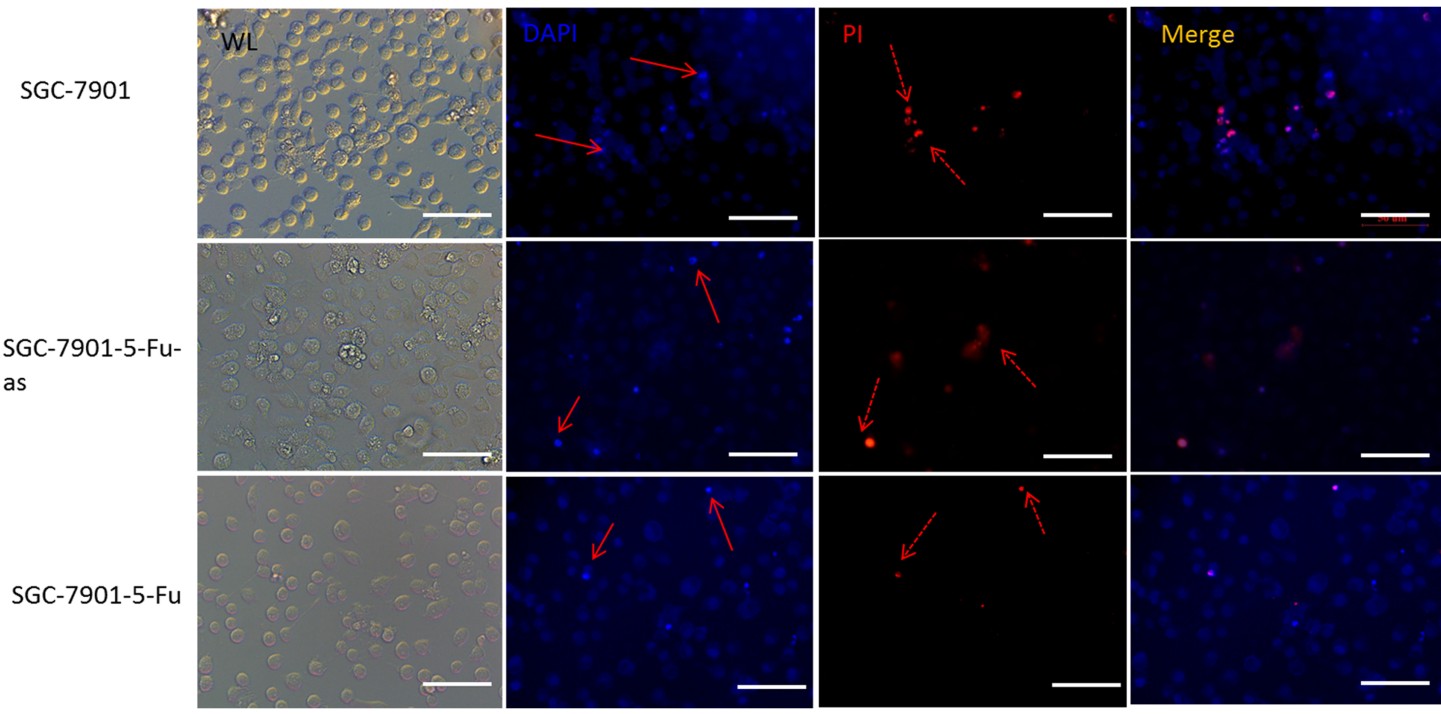

**Figure 5 The hsacirc_004413 knockdown promotes cell apoptosis and necrosis.** After transfection of hsacirc_004413 in SGC-7901-5-Fu and SGC-7901 cells, apoptosis and necrosis of the cells were detected by DAPI and PI staining. ML: white light, DAPI: 4′,6-diamidino-2-phenylindole, PI: Propidium Iodide, SGC-7901-5-Fu-as: antisense RNA for hsacirc_004413, red solid line arrow: apoptotic cells, dotted line arrow: necrotic cell.

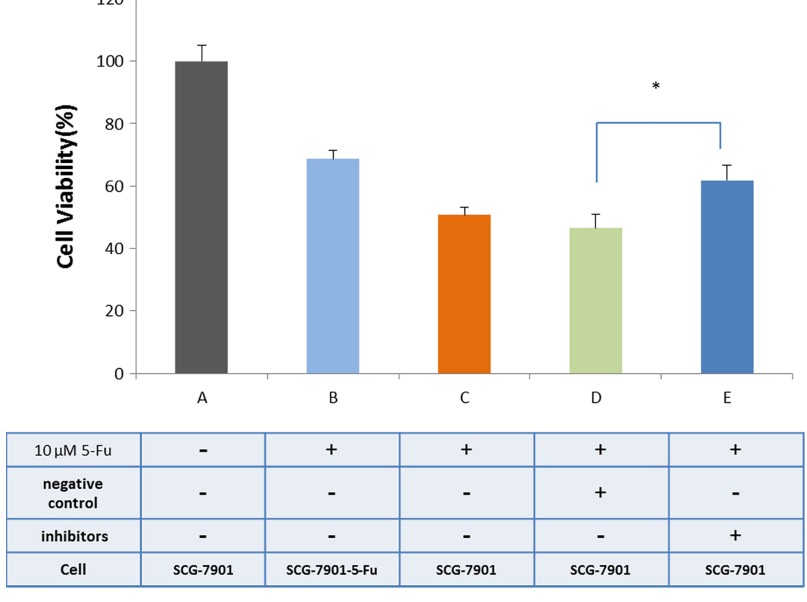

**Figure 6 Inhibition of miR-145-5p improved SGC-7901 cell survival.** The cells SGC-7901 that were sensitive to 5-Fu, were cultured under different culture conditions ((A) SCG-7901 cells; (B) SCG-7901-5-Fu cells + 10 μM 5-Fu; (C) SCG-7901 cells + 10 μM 5-Fu; (D) SCG-7901 cells + 10 μM 5-Fu + NC (negative control) inhibitors; (E) SCG-7901 cells + 10 μM 5-Fu + inhibitors.) for 24 h, and the cell survival rate was finally detected by CCK8. *$P < 0.05$ indicates significant differences.

thymidine, which are nucleosides needed for DNA replication. Cancer cells that divide rapidly lack a sufficient thymus to initiate the process of cell apoptosis (*Longley, Harkin & Johnston, 2003*).

However, damage to DNA of tumor cells caused by anti-tumor drugs can lead to drug resistance in tumor cells (*Seve & Dumontet, 2005*). We speculated that these upregulated circRNAs might be involved in the drug resistance mechanism of 5-FU in GC cells. The identities of 10 upregulated circRNAs were mapped using the circBase database (http://www.circbase.org/), and hsacirc_004413 was mapped to hsa_circ_0004650. The expression levels of hsacirc_004413 in sensitive and drug-resistant strains were investigated. The results showed that the expression levels of hsacirc_004413 in resistant strains were higher than those of the SCG-7901 and MNK-45 sensitive strains.

To further verify the role of hsacirc_004413 in drug resistance, we interfered with hsacirc_004413 in SCG-7901-5-Fu cells, and investigated the cell survival percentage. Compared with the control group, the cell survival percentage of the experimental group decreased. The results indicated that hsacirc_004413 knockout enhanced the sensitivity of drug resistant strains to 5-Fu. It also showed that HSACIRc_004413 was involved in the resistance of GC cells to 5-Fu. The 5-Fu activates apoptosis signals by inhibiting cell division (*Longley, Harkin & Johnston, 2003*; *Christensen et al., 2019*), and after apoptosis, the chromatin of cells is damaged (*Jin et al., 2010*). We also investigated apoptosis and necrosis. Some of the cells were stained with 4′,6-diamidino-2-phenylindole, resulting in very bright fluorescence, as shown in Fig. 5. These phenomena were more obvious in sensitive strains and resistant strains after knockdown. After propidium iodide staining, some cells showed red fluorescence, indicating that the cell membrane of these cells was incomplete, and cell necrosis occurred. The above phenomena were observed in both sensitive and resistant strains after knockdown. The results showed that the downregulated expression of hsacirc_004413 promoted the cells of drug resistant strains to become sensitive to 5-Fu. In this regard, we believe that hsacirc_004413 was involved in regulating the drug resistance of cells to 5-Fu in drug-resistant strains.

We used bioinformatics to predict microRNAs that may interact with hsacirc_004413 (Table S5). It has been reported that bioinformatics predicted that IRS1 and ACTB (-actin) mRNAs may be candidate targets of miR-145. Decreased IRS1 and ACTB expressions were detected by western blotting in 5-Fu resistant cells after downregulation of miR-145 (*Takagi et al., 2009*). Mir-145 inhibitor was therefore added to the medium of the sensitive SGC-7901 strain and cultured for 48 h with 10 µM 5-Fu. The results showed that compared with the vehicle group, the survival percentage of cells was significantly increased, indicating that the expression of miR-145 was downregulated, and the cells developed resistance to 5-Fu. Based on these results, we suggest that hsacirc_004413 adsorbed miR-145, regulated downstream gene expression, and caused cell resistance to 5-Fu.

## CONCLUSIONS

The hSACirC_004413 was detected in the SGC-7901-5-Fu cell line and verified by PCR. Bioinformatics were used to predict that mir-145-5p might bind with this component.

The hSACirC_004413 knockdown enhanced 5-Fu sensitivity in SGC-7901-5-FU cells and the percentages of cell apoptosis and necrosis significantly. Inhibition of Mir-145-5p improved the resistance of GC cells to 5-Fu. Based on the above results, we suggested that hsacirc_004413 inhibited apoptosis by sponging miR-145-5p, thereby promoting SGC-7901-5-Fu cell proliferation, and chemoresistance. These findings provided new insights into the mechanism of 5-Fu resistant gastric cancer, which is an urgent clinical problem to be solved.

### Funding
This work was supported by the Scientific Research Project of Shanghai Municipal Health and Family Planning Commission (Grant number 201640310). The funders had no role in study design, data collection and analysis, decision to publish, or preparation of the manuscript.

### Grant Disclosures
The following grant information was disclosed by the authors:
Scientific Research Project of Shanghai Municipal Health and Family Planning Commission: 201640310.

### Competing Interests
The authors declare that they have no competing interests.

### Author Contributions
- Fusheng Zhou conceived and designed the experiments, performed the experiments, analyzed the data, prepared figures and/or tables, and approved the final draft.
- Weiqun Ding conceived and designed the experiments, performed the experiments, analyzed the data, prepared figures and/or tables, and approved the final draft.
- Qiqi Mao performed the experiments, prepared figures and/or tables, and approved the final draft.
- Xiaoyun Jiang performed the experiments, prepared figures and/or tables, and approved the final draft.
- Jiajie Chen performed the experiments, prepared figures and/or tables, and approved the final draft.
- Xianguang Zhao performed the experiments, authored or reviewed drafts of the paper, and approved the final draft.
- Weijia Xu performed the experiments, authored or reviewed drafts of the paper, and approved the final draft.
- Jiaxin Huang performed the experiments, authored or reviewed drafts of the paper, and approved the final draft.
- Liang Zhong performed the experiments, analyzed the data, authored or reviewed drafts of the paper, and approved the final draft.

- Xu Sun conceived and designed the experiments, analyzed the data, authored or reviewed drafts of the paper, and approved the final draft.

## Data Availability

The data is available at Ensembl: SourceGene: ENSG00000179715; circBase_ID: hsa_circ_0004650.

http://circbase.org/cgi-bin/singlerecord.cgi?id=hsa_circ_0004650.

The data used to support the findings of this study are available at Dryad: https://datadryad.org/stash/dataset/doi:10.5061/dryad.v15dv41wv.

## Supplemental Information

Supplemental information for this article can be found online at http://dx.doi.org/10.7717/peerj.12629#supplemental-information.

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
