# Peer review of "The regulation of hsacirc_004413 promotes proliferation and drug resistance of gastric cancer cells by acting as a competing endogenous RNA for miR-145-5p"

_PeerJ, doi:10.7717/peerj.12629_

## Round 0.1 · original submission · Minor Revisions

The authors need to address all suggestions demanded by reviewers.

·

Basic reporting

The English used in this article is clear and technically correct. It would be advisable to check punctuation marks, as well as when capitalised and lowercase words are used throughout the text.
The references are correct. In line 317, the reference should keep the same format as the rest.
The estructure conforms to Peer J standards.
Figures are relevant and well labelled.
Some of the original supplementary information is in Chinese.
The hypothesis is relevant.

Experimental design

Some methodological aspects should be further elaborated in the manuscript to facilitate the understanding of the work.

Validity of the findings

The authors make a single conclusion from their work, but without showing its scientific relevance in the conclusion section.

Additional comments

Your introduction needs more detail.
You should update the treatment options for gastric cancer patients (ASCO Annual Meeting 2021) and explain that 5-Fu still has an important role in the treatment of gastric cancer patients today, so your research is clinically relevant.
In the introduction, regarding cicrRNA, I suggest that you improve the description of what it is and how it acts. You should provide more justification for your study (specifically, you should expand upon the knowledge about cicrRNA and drug resistance and gastric cancer).
In Line 51: made a gastric cancer cell line from SGC-7901 and not from MKN-45. You should explain the reason for your choice.
In the Results section:
1. After normalization, a total of 4,496 circRNA targets were found in SGC-7901 and 31 distinct circRNAs were selected because their expression. It would be good to explain it in more detail.
2. Hsacirc-004413 was highly expressed in 5-Fu resistant SGC-7901-5-Fu cells. In this experiment you used SGC-7901, SGC-7901-5-Fu and MKN-45 line cells. It would be advisable to explain why the MKN-45 cell line was used. As well, it would be very important for you to detail why you chose hsacirc-004413 and not another circRNA.
3. Line 167 should be in bold.
4. It should better explain the relationship between hsacirc-004413 and miR-145-5p and
5. It should better explain the relationship between hsacirc-004413 and miR-145-5p and how it arrives at the hypothesis that the resistance of hsacirc-004413 to 5-Fu may occur though the adsorption of miR-145-5p (Lines 178-181).

The Discussion is very extensive and some results do not appear in the results section, for example:
- Lines 203-208: hsacirc_004413 was compared with hsacirc_0004650.
- Lines 226-236: although this is discussed in more detail in the appendix, discussing this in the results section would make it easier to understand the importance of miR-145 in this context.

The relevance of the only conclusion of this work should be explained.

Reviewer 2 ·

Basic reporting

The idea of paper is well thought and data make justice to the Hypothesis.
The literature reported in MS is sufficient for the project. MS overall need better use of professional English language. I recommend to improve the language of result and discussion section. The article followed the professional structure although figure legends were not well written.

Experimental design

The experimental design of MS was comprehensive and well structure. It covered the scope of journal and finding mechanism of 5FU resistance in GC. The identification of miR-145-5p and circRNA was explained well for readers. The validation work was done with sound technical knowledge.

Validity of the findings

The work is novel for readers especially MS is very relevant to find 5FU resistance mechanism.

Additional comments

I strongly recommend to submit article with improve language and figure legend quality.

---

## Round 0.2 · accepted · Accept

Authors have incorporated all required changes.

Reviewer 2 ·

Basic reporting

The MS is improved from last submission.

Experimental design

The experimental design is rational and well plananed.

Validity of the findings

The findings will add to development of future treatment strategy